# Synergistic Effect of Propolis and Antibiotics on Uropathogenic *Escherichia coli*

**DOI:** 10.3390/antibiotics9110739

**Published:** 2020-10-27

**Authors:** Jean-Philippe Lavigne, Jérémy Ranfaing, Catherine Dunyach-Rémy, Albert Sotto

**Affiliations:** 1Virulence Bactérienne et Maladies Infectieuses, INSERM U1047, Université de Montpellier, Service de Microbiologie et Hygiène Hospitalière, CHU Nîmes, 30029 Nîmes, France; catherine.remy@chu-nimes.fr; 2Virulence Bactérienne et Maladies Infectieuses, INSERM U1047, Université de Montpellier, 30908 Nîmes, France; jeremy.ranfaing@gmail.com; 3Virulence Bactérienne et Maladies Infectieuses, INSERM U1047, Université de Montpellier, Service des Maladies Infectieuses et Tropicales, CHU Nîmes, 30029 Nîmes, France; albert.sotto@chu-nimes.fr

**Keywords:** antibiotics, bactericide curve, *Escherichia coli*, MIC, propolis, urinary tract infections

## Abstract

Urinary tract infections (UTIs) are the most common bacterial infections around the world. Uropathogenic *Escherichia coli* (UPEC) is among the main pathogens isolated in UTIs. The rate of UPEC with high resistance towards antibiotics and multidrug-resistant bacteria have increased dramatically and conduct to the difficulty to treat UTIs. Due to the rarefaction of new antibiotics molecules, new alternative strategies must be evaluated. Since many years, propolis has demonstrated an interesting antibacterial activity against *E. coli*. Here, we evaluated its activity added to antibiotics on a panel of UPEC with different resistance mechanisms. Minimal inhibitory concentrations (MICs) and time–kill curves of fosfomycin, ceftriaxone, ertapenem and ofloxacin, with and without propolis, were determined. Significant diminution of the MICs was observed using ceftriaxone or ofloxacin + propolis. Propolis alone had a bacteriostatic activity with time-dependent effect against UPEC. The addition of this nutraceutical improved the effect of all the antibiotics evaluated (except fosfomycin) and showed a synergistic bactericidal effect (fractional inhibitory concentrations index ≤ 0.5 and a decrease ≥ 2 log CFU/mL for the combination of propolis plus antibiotics compared with the antibiotic alone). Propolis is able to restore in vitro antibiotic susceptibility when added to antibiotics against UPEC. This study showed that propolis could enhance the efficiency of antibiotics used in UTIs and could represent an alternative solution.

## 1. Introduction

Urinary tract infections (UTIs) are common bacterial infections affecting a large proportion of women and around 20–40% of UTIs become recurrent [1,2,3,4]. The most frequently isolated organism during this infection is uropathogenic *Escherichia coli* (UPEC) [5]. Antimicrobial agents remain the main treatment. Among the different managements of UTIs, long-term low dose antibiotic use is currently the keystone of the preventive treatment for UTI recurrence. Indeed, prophylactic antibiotics have been shown to decrease UTI recurrence by 85% compared to patients with placebo [6]. Moreover, it has been suggested that the weekly cycling of antibiotics could be the most optimal preventative strategy [7,8]. However, all these protocols promoting prolonged antibiotic use often result in the emergence of multidrug-resistant (MDR) organisms [9] and increase the cost of care. In UTIs, the rate of MDR dramatically increased in these last years [10]. One of the main global threats is the widespread high-risk clones of extended spectrum beta-lactamase (ESBLs)-producing *E. coli* and particularly the pandemic B2-ST131 clone [11,12]. Furthermore, these pathogens have often also co-resistance to other important antibiotic groups like fluoroquinolones [1,13,14], limiting the therapeutic arsenal to treat these infections. The worldwide dissemination of MDR pathogens has severely reduced the efficacy of our antibiotic arsenal and increased the frequency of therapeutic failure. Consequently, the development of new therapeutic options to treat UTIs are needed. 

Recently, the interest of non-antibiotic alternative therapeutics has increased [15]. Among these options, nutraceuticals are promising. They consist of all the foods or food products which provide medical benefits and can be delivered under medical form [15]. Among them, propolis is a resinous material collected by bees from the exudates and buds of the plants and mixed with wax and bee enzymes [16]. Propolis has notably demonstrated an antimicrobial effect and it could improve the effect of other antimicrobial molecules like antibiotics [17,18,19]. Our team observed that propolis potentiated the effect of cranberry type-A proanthocyanidins (components of cranberry (*Vaccinium macrocarpon* Ait.) used in the prophylactic approach of a recurrent UTI) on different pathways of virulence of UPEC [20,21]. A recent randomized controlled trial using this association of nutraceuticals in 85 women with recurrent UTIs showed a reduction in the number of cystitis events in the first three months in the propolis and cranberry group after the adjustment of water consumption (0.7 versus 1.3, *p* = 0.02) [22].

The aim of the present study was to characterize the bactericidal effect of propolis on a panel of UPEC harboring different resistance mechanisms profiles and evaluate the synergy of this nutraceutical with the antibiotics usually used to treat UTIs.

## 2. Results

### 2.1. Determination of MICs

All the minimal inhibitory concentration (MIC) values are presented in Table 1 and Appendix A. The MIC values of the different antibiotics showed that the three antibiotics tested (ofloxacin (OFX), ceftriaxone (CRO), and ertapenem (ETP)) were active against NECS892841 and NECS30990, but NCS892841 were resistant to fosfomycin (FOS) (512 mg/L). NECS858785 and NECS864598 were resistant to OFX (MIC = 2 and 32 mg/L, respectively) and also to FOS (MIC = 128 and 512 mg/L, respectively). Finally, the two ESBL-producing UPEC (NECS892420 and NECS118564) were susceptible to ETP and FOS. All these data confirmed the expected resistant profiles of the different strains used in the study.

Secondly, we evaluated the addition of propolis at 0.5 × and 1 × MIC to the different antibiotics. A very low variation of MICs was observed with propolis at 0.5 × MIC (Appendix A). The results obtained with the addition of propolis at 1 × MIC showed a more pronounced effect of the combination with a decrease in almost all of the MICs (Table 1). The decrease was significant with the combination of CRO and propolis for 5/7 tested strains (NECS892841 NECS858785, NECS864598, NECS892420 and NECS118564), with the combination of OFX and propolis for 3/7 tested strains (CFT073, NECS892420 and NECS118564), and with the combination of ETP and propolis for 1/7 tested strains (CFT073). The combination of FOS and propolis had no clear effect on the antibiotic activity.

The effect of the combination of antibiotics and propolis was determined following European Committee on Antimicrobial Susceptibility Testing (EUCAST) recommendations [24]. The results are presented in Table 2.

We observed that propolis plus OFX was the combination that showed synergistic activity against all the strains tested. A same trend was observed to propolis plus CRO and ETP except for one strain (NECS864598 and NECS892841, respectively) which presented an additive effect. Lastly, no synergistic effect was observed with the combination of propolis and FOS. 

### 2.2. Time–Kill Curves of Propolis and Antibiotics Alone

Propolis alone at different increasing concentrations was bacteriostatic from 1 h to 24 h against the seven strains tested (Figure 1). It exhibited a time-dependent antimicrobial activity.

### 2.3. Time–Kill Curves of a Combination of Propolis and Antibiotics

The bactericidal activity of the propolis and the antibiotics alone is shown in Table 3. ETP demonstrated bactericidal activity against all the studied strains. CRO, FOS and OFX were bactericidal against five, four and three strains, respectively. Propolis alone showed no bactericidal effect against any strain. 

The in vitro activity of the propolis in combination with antibiotics is shown in Table 4. Propolis plus ETP was the combination that showed synergistic activity against all the strains tested. Propolis plus OFX demonstrated the more prolonged synergistic effect against NECS30990 with an activity detected at 24 h (Figure 2). Propolis plus CRO showed a synergistic activity against UPEC strains except the ESBL producers. Finally, propolis plus FOS did not present synergy against any strain. A slight bactericidal activity was observed at 6 h against NECS30990 (Figure 2).

The activity of propolis at 1 × MIC in combination with antibiotics against NECS30990 (the strain the most susceptible to these combinations) is shown in Figure 2. If a clear bactericidal and synergistic effect was noted with OFX, CRO and ETP, propolis did not have any activity in combination with FOS.

### 2.4. In Vitro Selection of Resistant Mutants

Any resistant mutants were detected with propolis alone or in combination with antibiotics.

## 3. Discussion

The aim of this study was to confirm the in vitro antimicrobial effect of propolis against a panel of clinical UPEC and to determine if this product could improve the efficacy of antibiotics used to treat UTIs. Indeed, the antibacterial activity of propolis has been evaluated over many years and *E. coli* was the preferentially tested pathogen [19]. The previous MIC values determined varied between 116 to approx. 5000 mg/L [19]. Our data were in accordance with the lowest MIC values observed (128–256 mg/L) and confirmed the effect of propolis. Previous works have determined that the activity of propolis depends on the chemical composition of this nutraceutical and varies between the different origins of the propolis. Among these products, raw propolis (used in our work), extracted by ethanol, released the most active ingredients and demonstrated the better antimicrobial activity [26,27,28]. We used this protocol in our experiments to obtain the higher concentrations of phenolic compounds. This certainly explains the important activity noted in our study. The main active groups of chemical compounds found in propolis are now well characterized and polyphenols (e.g., flavonoids and phenolic acids) present the most important antibacterial activity [26,27]. It is often considered that this antimicrobial activity of these compounds is due to both a direct action on the microorganism and a stimulation of the immune system activating natural defences of the organism [16]. In the direct action on bacteria, the propolis interferes on the permeability of their membrane, disrupts the membrane potential and ATP production, and decreases bacterial mobility [19,29].

Our study also determined that propolis had a bacteriostatic activity with a time-dependent effect against UPEC, a result not previously noted. Moreover, if propolis had no significant bactericidal activity against our strains, it presented a strong in vitro activity combined with different antimicrobials agents. Thus, we noted that, at a concentration of 1 × MIC, propolis decreased the MIC values of the different antibiotics tested against all the clinical UPEC studied. Even if the fluoroquinolones-resistant and ESBL-producing strains did not recover susceptibility against ofloxacin or ceftriaxone, respectively, we observed an unexpected significant decrease in the resistance level closed to their susceptibility thresholds (Table 2 and Table 3).

Interestingly, we observed that the combination of propolis with the different antibiotics tested potentiates the effect of these molecules alone against the clinical UPEC strains. This suggests that the use of propolis could be added in the treatment of UTIs. Recently, we observed by a transcriptomic approach that propolis potentiated the effect of cranberry proanthocyanidins (well characterized inhibitors of Type-I fimbriae *E. coli* adhesion [30]) on adhesion, motility (swarming and swimming), biofilm formation (early formation and fully formed biofilm), iron metabolism and the stress response of UPEC [20]. Moreover, we observed that this effect was effective against all the UPEC whatever their origin or virulence profile [21]. The combination of antibiotics and propolis harboring an interesting action on UPEC virulence could represent a new strategy due to the action of propolis in addition to antibiotics. It is noteworthy that synergistic activity was observed when propolis was combined with CRO, OFX and ETP, but we did not observe a same trend concerning the combination with FOS, the first antibiotic of choice to treat the non-complicated UTIs of community origin [31,32]. However, the resistance to this antibiotic was particularly low [33], notably among ESBL-producing uropathogenic enterobacteria [34], suggesting that the absence of activity of propolis was less important in this situation. 

Propolis plus β-lactams was the combination that showed the best synergism results against more of the tested strains (five to seven out of seven). This finding was previously noted: a synergistic antibacterial activity was observed with the combination of β-lactams and apigenin (a flavonoid compound of propolis) against methicillin-resistant *Staphylococcus aureus* [35] and ceftazidime and apigenin against ceftazidime-resistant *Enterobacter cloacae* [36]. Moreover, this effect was confirmed in in vivo BALB/c mice model using propolis plus cefixime to treat *Salmonella* infection [37]. Due to the worldwide diffusion of ESBL-producing *E. coli* [11], the addition of propolis to antibiotics could be evaluated to propose a new alternative for UTI treatment. We could also note that propolis plus ofloxacin was particularly synergistic against susceptible UPEC. Even if a rise of fluoroquinolone-resistant strains has been observed [38,39], this antibiotic remains a first-line antibiotic treatment for UTIs and its association with propolis could be interesting. Our study highlights two main questions: (i) can we observed the synergistic effect described in our in vitro model in the urines of patients? Indeed, it is well known that MICs and Minimal Bactericidal Concentrations are medium depending [40] and urines are a complex environment [41]; (ii) moreover, how to obtain these high quantities (128–256 mg/L) of propolis needed to obtain the antibacterial effect in vivo. Indeed, as we previously observed, different active compounds from propolis are well present in urines but in very low concentration [42]. However, Kalia et al. [37] observed a synergistic effect after the administration of 150 to 225 mg/kg of propolis in combination with cefixime (2 mg/kg) in an in vivo BALB/c mice model. In vivo studies will be needed to definitively conclude on the synergy of propolis in combination with antibiotics in urines.

With the dramatic increase in antibiotic resistance since this century [11], it is essential to develop new strategies to prevent or treat UTIs. In this context, nutraceuticals could represent an alternative and/or a complement to antibiotics. Here, we observed a clear synergistic effect between propolis and some antibiotics used in UTI treatment, suggesting that propolis may be an alternative for this treatment. To investigate further the possible usefulness of this compound, new data from pharmacokinetics and pharmacodynamics and in vivo efficiency in experimental models of infections are required.

## 4. Materials and Methods 

### 4.1. Bacterial Strains

Six clinical strains isolated from different origins (cystitis and pyelonephritis) were used: NECS892841 and NECS30090 (wild strains that did not produce β-lactamases), NECS858785 and NECS864598 (strains with only (fluoro)-quinolone resistance), NEC892420 and NEC118564 (CTX-M-15 and TEM-24-producing strains respectively) (Table 5). A reference strain was also studied CFT073.

### 4.2. Compounds

Mueller–Hinton (MH) growth medium (Invitrogen, Villebon sur Yvette, France) were used for all the experiments as recommended by EUCAST [23]. Plates of MH agar were used for the colony counts. 

The propolis extract (Plantex, Sainte-Geneviève-des-Bois, France) used in this study is a hydroalcoholic extract of blended propolis from various origins mixed with carob in a proportion of (60/40, *w*/*w*). The propolis was characterized by HPLC showing 2% of galangin (Lot 38123). It was extracted by ethanol, diluted in 50 mL of PBS and incubated at 37 °C by shaking at 100 rpm for 8 h. Then, the solution was clarified by centrifugation (4000 rpm, 20 °C, 10 min). Supernatant was sterilized by filtration.

All the drugs tested were used as standard laboratory powders (Sigma-Aldrich, Saint-Quentin Fallavier, France): OFX, CRO, ETP and FOS. The stock solutions from the dry powders were prepared at a concentration of 5120 mg/L according to the Clinical and Laboratory Standards Institute (CLSI) recommendations [43].

### 4.3. Determination of the Minimal Inhibitory Concentration (MIC)

MICs were determined using the broth microdilution technique according to the EUCAST guideline [23]. Serial two-fold dilutions ranging from 5196 to 0.012 mg/L were prepared in MH broth. Each well was inoculated with the overnight cultures of the bacteria at a final concentration of 5 × 10^5^ CFU/mL. The trays were covered and placed in a plastic box to prevent evaporation, and then incubated at 37 °C for 24 h. The MICs of these organisms to the antibiotic–propolis combination were determined for two fixed concentrations of propolis: 0.5× or 1 × MIC, concentrations compatible to the dose delivery by commercialized caps. The MIC was defined as the lowest concentration of antimicrobials that completely inhibited the bacterial growth. All assays were performed in triplicates on different days. Isolates were classified as susceptible, intermediately susceptible, or resistant, according to the interpretive criteria of EUCAST [23] for all the antibiotics.

The effect of the combination of antibiotics and propolis has been defined according to the fractional inhibitory concentration (FIC) index corresponding to the (MIC of drug A, tested in combination)/(MIC of drug A, tested alone) + (MIC of drug B, tested in combination)/(MIC of drug B, tested alone). A synergistic effect was obtained when the FIC index ≤0.5. An additive effect was observed when the FIC index was comprised between >0.5 and 1. Indifference was noted when the FIC index was comprised between >1 and <2 [24].

### 4.4. Time–Kill Assays

The concentrations used for the different antibiotics tested corresponded to the MICs obtained by microdilution according to National Committee for Clinical Laboratory Standards [44]. The assays were also performed at 1 × and 0.5 × MIC of propolis. Experiments were carried out with a starting inoculum of 10^5^ CFU/mL. The time–kill curves were elaborated by plotting the mean colony counts (log CFU/mL) of propolis alone and in combination with the antibiotics versus time. The bacterial suspensions of the isolates were incubated at 37 °C, with gentle shaking, and the variable counts were performed at different times (0, 1, 2, 3, 5, 6 and 24 h of incubation). One milliliter of the suspensions was withdrawn and serially diluted with a sterile saline solution. Then, 100 µL of the bacterial suspensions or dilutions were plated, and the CFU was determined after the overnight incubation of the plates at 37 °C. 

The lowest limit of detection for the time–kill assay was 1 log CFU/mL. The results were interpreted by the effect of the combination in comparison with that of the most active agent alone. Synergy was defined as a decrease ≥ 2 log CFU/mL for the drugs’ combination compared with the most active single agent. The bactericidal activity was defined as a decrease of ≥ 3 log CFU/mL from the initial inoculum, and the bacteriostatic effect was defined as no change with respect to the initial bacterial [25]. Experiments were performed three times on separate occasions. 

### 4.5. In Vitro Selection of Resistant Mutants

For the detection of resistant mutants, the MICs of antibiotics and propolis alone or in combination against the studied strains were carried out in triplicate for five colonies obtained at each time-point during the time–kill assays. 

### 4.6. Statistical Analysis

For the MIC experiments, we assumed that a difference of 3 dilutions or more between 2 conditions was considered a significant result.

## Figures and Tables

**Figure 1 antibiotics-09-00739-f001:**
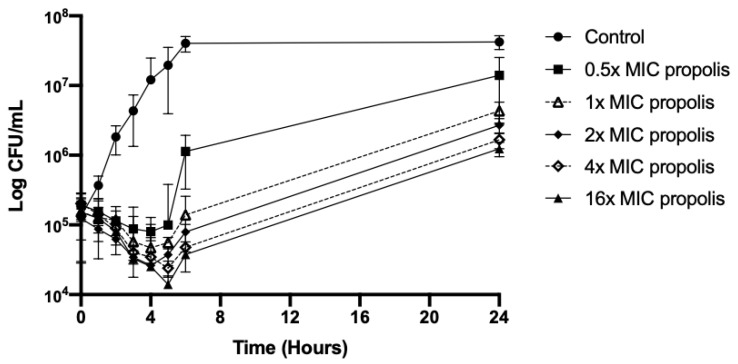
Time–kill curves for propolis alone at different concentrations against the clinical strain NECS30990.

**Figure 2 antibiotics-09-00739-f002:**
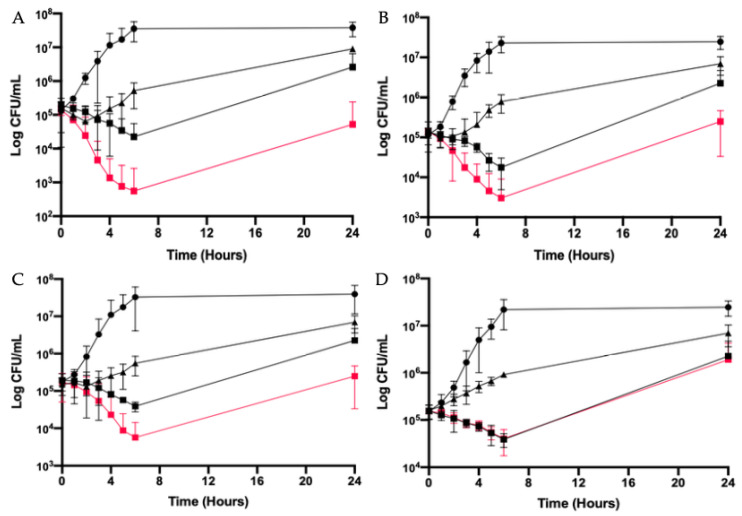
Time–kill curves for propolis at 1 × MIC in combination with (**A**), ofloxacin; (**B**), ceftriaxone; (**C**), ertapenem and (**D**), fosfomycin against the clinical strain NECS30990. Filled circles, growth control; filled triangles, antibiotics (1 × MIC); filled squares, propolis (1 × MIC); red square, combination of propolis (1 × MIC) + antibiotic (1 × MIC).

**Table 1 antibiotics-09-00739-t001:** Minimal inhibitory concentrations (MICs) of antibiotics, propolis (1 × MIC) and both for a panel of uropathogenic *Escherichia coli* (UPEC). All the data are determined in mg/L. The different thresholds were: ofloxacin (OFX), R > 0.5 mg/L; ceftriaxone (CRO), R > 2 mg/L; ertapenem (ETP), R > 1 mg/L; fosfomycin (FOS), R > 32 mg/L) [23].

Strains	Resistance	OFX	OFX + Propolis	CRO	CRO + Propolis	ETP	ETP + Propolis	FOS	FOS + Propolis	Propolis
CFT073	Sensitive	0.25	*0.06 * ^1^	0.5	0.125	0.06	*0.006*	8	4	256
NECS892841	Sensitive	0.5	0.125	1	*0.125*	0.06	0.03	**512 ^2^**	**128**	256
NECS30990	Sensitive	0.25	0.06	0.5	0.06	0.03	0.012	8	8	128
NECS858785	OFX R	**2**	**0.5**	0.25	*<0.06*	0.06	0.012	**128**	**64**	256
NECS864598	OFX R	**32**	**8**	0.5	*0.06*	0.125	0.06	**512**	**256**	256
NECS892420	ESBL	**>32**	***8***	**>32**	***2***	0.25	0.06	8	8	256
NECS118564	ESBL	**>32**	***8***	**>32**	***2***	0.5	0.125	8	4	256

^1^ In italic and underlined are the results with significant difference between the experiments with and without propolis; ^2^ In bold, the results corresponding to a resistant profile. ESBL, extended-spectrum β-lactamase

**Table 2 antibiotics-09-00739-t002:** Fractional inhibitory concentrations (FICs) and the FIC index of the combination of propolis and antibiotics against a panel of UPEC. The FIC index is the sum of FICs calculated. A synergistic effect was obtained when the FIC index ≤ 0.5. An additive effect was observed when the FIC index was comprised between > 0.5 and 1. Indifference was noted when the FIC index was comprised between > 1 and < 2 [24].

Strains	FIC OFX	FIC Propolis	FIC Index	FIC CRO	FIC Propolis	FIC Index	FIC ETP	FIC Propolis	FIC Index	FIC FOS	FIC Propolis	FIC index
CFT073	0.24	2.3 × 10^−5^	**0.25**	0.25	4.9 × 10^−4^	**0.26**	0.1	2.3 × 10^−5^	**0.11**	0.5	0.016	*0.51*
NECS892841	0.25	4.9 × 10^−4^	**0.26**	0.125	4.9 × 10^−4^	**0.13**	0.5	1.2 × 10^−4^	*0.51*	0.25	**0.5**	*0.75*
NECS30990	0.24	2.3 × 10^−5^	**0.25**	0.12	4.7 × 10^−4^	**0.13**	0.4	9.4 × 10^−5^	**0.41**	1	0.063	1.06
NECS858785	0.25	0.002	**0.26**	0.25	*1.2* × *10^−4^*	**0.26**	0.2	4.7 × 10^−5^	**0.21**	0.5	**0.25**	*0.75*
NECS864598	0.25	0.03125	**0.28**	0.5	4.9 × 10^−4^	*0.51*	0.48	2.3 × 10^−4^	**0.49**	0.5	**1**	1.5
NECS892420	0.13	0.03125	**0.16**	0.031	0.008	**0.04**	0.24	2.3 × 10^−4^	**0.25**	1	0.031	1.03
NECS118564	0.06	0.03125	**0.10**	0.031	0.008	**0.04**	0.25	4.8 × 10^−4^	**0.26**	0.5	0.016	*0.52*

In bold, synergism effect; in italic, additive effect. OFX, ofloxacin; CRO, ceftriaxone; ETP, ertapenem; FOS, fosfomycin

**Table 3 antibiotics-09-00739-t003:** Bactericidal activity of antibiotics and propolis alone against a panel of UPEC [25].

Strains	OFX	CRO	ETP	FOS	Propolis
CFT073	B (2–24 h)	B (2–24 h)	B (2–24 h)	B (2–24 h)	-
NECS892841	B (2–24 h)	B (3–24 h)	B (2–24 h)	-	-
NECS30990	B (2–24 h)	B (2–24 h)	B (2–24 h)	B (2–24 h)	-
NECS858785	-	B (2–24 h)	B (2–24 h)	-	-
NECS864598	-	B (2–24 h)	B (2–24 h)	-	-
NECS892420	-	-	B (2–24 h)	B (3–24 h)	-
NECS118564	-	-	B (2–24 h)	B (4–24 h)	-

OFX, ofloxacin; CRO, ceftriaxone; ETP, ertapenem; FOS, fosfomycin; B, bactericidal; -, no bactericidal activity found; (), time frame in hours of the in vitro activity found.

**Table 4 antibiotics-09-00739-t004:** Bactericidal activity of propolis (1 × MIC) in combination with antibiotics against a panel of UPEC [25].

Strains	OFX + PRO	CRO + PRO	ETP + PRO	FOS + PRO
CFT073	B + S (4–6 h)	B + S (4–5 h)	B + S (4–6 h)	-
NECS892841	B + S (4–6 h)	B + S (4–6 h)	B + S (5 h)	-
NECS30990	B + S (4–24 h)	B + S (4–6 h)	B + S (5–6 h)	B (6 h)
NECS858785	-	B + S (4–6 h)	B + S (4–6 h)	-
NECS864598	-	B + S (4–6 h)	B + S (5–6 h)	-
NECS892420	-	-	B + S (5–6 h)	-
NECS118564	-	-	B +S (4–6 h)	-

OFX, ofloxacin; CRO, ceftriaxone; ETP, ertapenem; FOS, fosfomycin; PRO, propolis; B, bactericidal; S, synergistic; -, in vitro activity found; (), time frame in hours of the in vitro activity found.

**Table 5 antibiotics-09-00739-t005:** Main characteristics of the *Escherichia coli* isolates used in this study.

Strain	Source	Resistance Profile ^1^	Phylogroup	Serogroup	ST
CFT073	Blood	-	B2	O6	73
NECS892841	Urine (cystitis)	-	B2	O18	95
NECS30090	Urine (pyelonephritis)	-	B2	O9	10
NECS858785	Urine (cystitis)	NAL, OFX	B2	O6	73
NECS864598	Urine (pyelonephritis)	NAL, OFX, CIP	B2	O21	12
NEC892420	Urine (cystitis)	AMX, AMC, TIC, TCC, PIP, TZP, CTX, CAZ, KAN, TOB, GEN, NET, NAL, OFX	B2	O6	127
NEC118564	Urine (pyelonephritis)	AMX, AMC, TIC, TCC, PIP, TZP, CTX, CAZ, KAN, TOB, GEN, NET, NAL, OFX, CIP, SXT	B2	O25	131

^1^ AMX, amoxicillin; AMC, amoxicillin/clavulanic acid; CAZ, ceftazidime; CIP, ciprofloxacin; CTX, cefotaxime; GEN, gentamicin; KAN, kanamycin; NAL, nalidixic acid; NET, netilmicin; OFX, ofloxacin; PIP, piperacillin; SXT, cotrimoxazole; TIC, ticarcillin; TCC, ticarcillin/clavulanic acid; TOB, tobramycin; TZP, piperacillin/tazobactam.

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
