# Peer review of "Synergistic Effect of Propolis and Antibiotics on Uropathogenic Escherichia coli"

_antibiotics, 2020, doi:10.3390/antibiotics9110739_

Round 1

Reviewer 1 Report

This is an interesting study investigating the synergistic effect of propolis and antibiotics on UPEC strains (different number of strains showed „synergy“ according to the antibiotics tested). The authors found a significant reduction of the MIC/MBC of the tested antibiotics in combination with propolis, which is bacteriostatic with a MIC of about 256 mg/l in their study.

The study is well performed and presented. The authors also discussed correctly that the urinary pharmacokinetic of propolis needs to be determined better, before therapeutic or prophylactic studies could be performed, because the concentrations in urine are not known. In addition it is not known, whether in the medium „urine“ also a synergistic effect can be found, because MIC and MBC are also medium depending. I suggest that this should also be mentioned in this chapter.

One question remains: why did the investigators not use the checkerboard technique to determine the synergistic effect. As I understood they only have used the MIC concentration of propolis and determined the change of the MIC of the antibiotic tested. Using the checkerboard technique the investigators may have detected that MIC concentrations of propolis even lower than half MIC could have been synergistic with the antibiotics. Synergy should be defined  according to the fractional inhibitory concentration (FIC) index and fractional bactericidal concentration (FBC) index:  (MIC of drug A, tested in combination)/(MIC of drug A, tested alone) + (MIC of drug B, tested in combination)/(MIC of drug B, tested alone). FBC calculation is carried out accordingly.

Then there are commonly accepted definitions:

  1. Synergy; FIC/ FBC ≤ 0.5
  2. Indifference; FIC/ FBC >0.5 to <4.0
  3. Antagonism; FIC/FBC ≥4.0

At least the authors should comment on that, why they did not use the commonly accepted definitions.

Nevertheless, such kind of studies are important to find ways either to keep or restore antimicrobial activity of the current antibiotics.

Author Response

This is an interesting study investigating the synergistic effect of propolis and antibiotics on UPEC strains (different number of strains showed „synergy“ according to the antibiotics tested). The authors found a significant reduction of the MIC/MBC of the tested antibiotics in combination with propolis, which is bacteriostatic with a MIC of about 256 mg/l in their study. The study is well performed and presented. The authors also discussed correctly that the urinary pharmacokinetic of propolis needs to be determined better, before therapeutic or prophylactic studies could be performed, because the concentrations in urine are not known. In addition it is not known, whether in the medium „urine“ also a synergistic effect can be found, because MIC and MBC are also medium depending. I suggest that this should also be mentioned in this chapter.

We thank the reviewer for these positive comments. We believe that we have improved the manuscript thanks to his/her comments.

We added the notion of the influence of the medium ‘urine’ on MIC and MBC.

One question remains: why did the investigators not use the checkerboard technique to determine the synergistic effect. As I understood they only have used the MIC concentration of propolis and determined the change of the MIC of the antibiotic tested. Using the checkerboard technique the investigators may have detected that MIC concentrations of propolis even lower than half MIC could have been synergistic with the antibiotics. Synergy should be defined  according to the fractional inhibitory concentration (FIC) index and fractional bactericidal concentration (FBC) index:  (MIC of drug A, tested in combination)/(MIC of drug A, tested alone) + (MIC of drug B, tested in combination)/(MIC of drug B, tested alone). FBC calculation is carried out accordingly.

Then there are commonly accepted definitions:

  1. Synergy; FIC/ FBC ≤ 0.5
  2. Indifference; FIC/ FBC >0.5 to <4.0
  3. Antagonism; FIC/FBC ≥4.0

At least the authors should comment on that, why they did not use the commonly accepted definitions.

We calculated the FIC and FIC index for all the antibiotics in combination with propolis accordingly to the EUCAST methodology. A new Table 2 presents these results.

In the first version of our manuscript, synergy was defined as a decrease ≥ 2 log CFU/mL for the drugs combination compared with the most active single agent. The bactericidal activity was defined as a decrease ≥ 3 log CFU/mL from the initial inoculum, bacteriostatic effect was defined as no change respect to the initial bacterial. This synergistic effect was calculated using time-kill curves.

Reviewer 2 Report

To the authors

This is an interesting topic – Urinary tract infections (UTI’s) are an important infection globally and inappropriate treatment is one of the main drivers of antibiotic resistance in bacteria.   Although the use of the natural product – propolis - for treatments has been known for decades and some of the mechanisms of action (the effects on OM of E.coli ) are recognised, this paper puts together some new data on the effect of the (synergistic) combinations of antibiotics with propolis against clinical strains and calls for more information on pharmacology of the propolis

I have found many sentences that are incomplete

Line

Abstract

ln 17 meaning?

Ln25 please qualify level of synergistic effect

Introduction

Ln59 unclear

Results

Why have you not carried out chequerboard analysis to determine activity in conventional way

Table 1 - Level of significance not stated and a confusing table

MBC determinations not included?

Table 2 difficult to understand without reference to methodology 

I could not locate table 3 although the authors refer to it pg  4   ln 98     ?

Ln 108 /109 refers to which table/figure ?

Discussion ln 152; 171  sense?

I was surprised that such a small number of isolates had been tested and needs reference strains with well defined resistance charaterised

Defined reductions /increases of activity are used to show synergy/additive or antagonism with MIC

tests – why have they not been used here?

Author Response

This is an interesting topic – Urinary tract infections (UTI’s) are an important infection globally and inappropriate treatment is one of the main drivers of antibiotic resistance in bacteria. Although the use of the natural product – propolis - for treatments has been known for decades and some of the mechanisms of action (the effects on OM of E.coli ) are recognised, this paper puts together some new data on the effect of the (synergistic) combinations of antibiotics with propolis against clinical strains and calls for more information on pharmacology of the propolis

We thank the reviewer for these positive comments. We believe that we have improved the manuscript thanks to his/her comments.

I have found many sentences that are incomplete

Line

Abstract 

-ln 17 meaning?

We rewrote the sentence in the new version of the manuscript.

-Ln25 please qualify level of synergistic effect

We added the information of the Fractional inhibitory concentrations index ≤0.5 and the decrease ≥ 2 log CFU/mL for the combination propolis plus antibiotics compared with the antibiotic alone.

Introduction 

-Ln59 unclear

We rewrote the sentence in the new version of the manuscript.

-Results 

Why have you not carried out chequerboard analysis to determine activity in conventional way

This analysis was added in the new version of the manuscript (Table 2).

-Table 1 - Level of significance not stated and a confusing table

The level of significance was stated in the Material and Method section (4.6). We proposed a slight modification in Table 1.

-MBC determinations not included?

The MBC was not strictly determined. However, the bactericidal effect has been presented wit the time kill curves (as recommended by EUCAST – Ref 25).

-Table 2 difficult to understand without reference to methodology 

We added a reference (Ref 26) in the Tables (new Tables 3 and 4).

-I could not locate table 3 although the authors refer to it pg  4  ln 98 ?

The Table was inserted in the main document. It corresponds to Table 4 in the new version of the manuscript.

-Ln 108 /109 refers to which table/figure ?

No Table/figure are linked to this paragraph. We moved paragraph to avoid confusion.

-Discussion ln 152; 171  sense?

We rewrote the sentences.

-I was surprised that such a small number of isolates had been tested and needs reference strains with well defined resistance charaterised

We agree with the reviewer that the panel is small. However, it corresponds to a representative panel of strains isolated from UTIs. We characterized these isolates to avoid a ”clonal effect” due to the presence of a same and unique clone among the 6 strains.

-Defined reductions /increases of activity are used to show synergy/additive or antagonism with MIC tests – why have they not been used here?

We added paragraphe and Table (new Table 2) in this new version of the manuscript.